# Detection of Rift Valley Fever Virus RNA in Formalin-Fixed Mosquitoes by In Situ Hybridization (RNAscope^®^)

**DOI:** 10.3390/v13061079

**Published:** 2021-06-05

**Authors:** Sarah Lumley, Laura Hunter, Kirsty Emery, Roger Hewson, Anthony R. Fooks, Daniel L. Horton, Nicholas Johnson

**Affiliations:** 1Microbiology Services Division, Public Health England, Wiltshire SP4 0JG, UK; slumley@mail.dstl.gov.uk (S.L.); laura.hunter@phe.gov.uk (L.H.); kirsty.emery@phe.gov.uk (K.E.); roger.hewson@phe.gov.uk (R.H.); 2School of Veterinary Medicine, University of Surrey, Guildford GU2 7XH, UK; d.horton@surrey.ac.uk; 3Virology Department, Animal and Plant Health Agency (Weybridge), Woodham Lane, Surrey KT15 3NB, UK; tony.fooks@apha.gov.uk

**Keywords:** Rift Valley fever virus, mosquito, virus detection

## Abstract

Rift Valley fever virus (RVFV) causes a zoonotic mosquito-borne haemorrhagic disease that emerges to produce rapid large-scale outbreaks in livestock within sub-Saharan Africa. A range of mosquito species in Africa have been shown to transmit RVFV, and recent studies have assessed whether temperate mosquito species are also capable of transmission. In order to support vector competence studies, the ability to visualize virus localization in mosquito cells and tissue would enhance the understanding of the infection process within the mosquito body. Here, the application of in situ hybridization utilizing RNAscope^®^ to detect RVFV infection within the mosquito species, *Culex pipiens,* derived from the United Kingdom was demonstrated. Extensive RVFV replication was detected in many tissues of the mosquito with the notable exception of the interior of ovarian follicles.

## 1. Introduction

Rift Valley fever is an economically significant zoonotic disease of livestock in sub-Saharan Africa. Outbreaks can occur suddenly, often driven by the onset of heavy rainfall, resulting in significant morbidity and mortality amongst livestock and humans [1]. The causative agent is Rift Valley fever virus (RVFV), classified within the family *Phenuiviridae* and genus *Phlebovirus*, which has a tri-segmented genome consisting of a small (S) segment encoding the virus nucleoprotein and a non-structural protein, a medium (M) segment encoding two glycoproteins, Gn and Gc, and a large (L) segment encoding the virus RNA-dependent RNA polymerase. Rift Valley fever virus is transmitted to and between mammals by mosquitoes, although human infections usually result from close contact with contaminated carcasses. Abortions in ruminants are often observed within livestock following infection with RVFV. In addition, the movement of infected livestock has been implicated in the establishment of RVFV in areas outside of sub-Saharan Africa, such as the Arabian Peninsula [2] and islands within the Indian Ocean [3,4].

A range of mosquito species have been associated with transmission of RVFV, particularly those within the genera *Aedes* and *Culex*, reviewed in [5,6]. Establishing the vector competence of particular species has been essential in understanding the epidemiology of RVFV transmission, both during outbreaks and inter-epidemic periods. Virus detection in mosquitoes usually relies on virus isolation in cell-cultures and reverse transcription polymerase chain reaction (RT-PCR). However, the ability to detect the virus in the context of organ structures, using immunohistochemistry, provides further information on the ability of arthropod-borne viruses to infect particular mosquito species [7,8]. Formalin fixation, whilst inactivating the virus for safe handling outside of primary containment, can limit the application of some techniques. RNAscope^®^, a form of in situ hybridization using RNA probes, was developed to detect RNA within formalin-fixed, paraffin-embedded tissue [9]. A recent report has developed RNAscope^®^ for detection of RVFV within mammalian liver tissue [10], a key tissue for diagnosis in livestock, using probes that target the virus polymerase gene on the L segment.

In this study, we have adapted a similar approach, but using probes designed against the virus nucleoprotein gene on the S segment, to investigate the presence and distribution of RVFV RNA within an infected mosquito. Control probes targeting mosquito glyceraldehyde-3-dehydrogenase (*GAPDH*) gene confirmed the methodology was effective for insect tissue. Staining with RVFV probes demonstrated extensive RVFV RNA labelling in key structures such as the basal layer of the mosquito midgut and the proventriculus, suggesting points of entry for the virus following consumption of a bloodmeal.

## 2. Materials and Methods

### 2.1. Mosquitoes, Viruses and Infection Protocol

A colonized, indigenous mosquito strain of *Culex pipiens*, Linnaeus 1758, was used as the infection model. This strain, designated Caldbeck [11], was originally trapped from a location in London (United Kingdom) in 2011. The RVFV strain used for infections was ZH501, originally isolated from a human with haemorrhagic fever in Egypt in 1977, and it was grown in Vero cells. This was diluted in horse blood to a final titre of 10^7^ plaque forming units (pfu)/mL. Mosquito infection with RVFV was conducted as previously described [12]. A second group was fed horse blood with tissue culture media as a negative control group. Blood-fed mosquitoes were maintained at 25 °C for 7 days prior to processing. Briefly, individual mosquitoes were dipped in 70% ethanol to reduce hydrophobicity and then immersed in 0.5 mL 10% buffered formalin within a screw-cap tube for 5 days at room temperature. Mosquitoes were paraffin embedded, sectioned (3 µm) and mounted on glass slides.

### 2.2. In Situ Hybridization: RNAscope^®^

RNAscope^®^ (Advanced Cell Diagnostic ACD) is a commercially available RNA in situ hybridization (RNA-ISH) technology, applied here to study RVFV replication and localization with light microscopy of the whole mosquito following oral infection. Novel probes targeting the negative sense nucleoprotein-coding gene of RVFV strain ZH501 (nucleotides 915–1652; GenBank accession number DQ380148.1) and mosquito GAPDH gene from *Culex quinquifasciatus*, Say 1823, (nucleotides 54–810; GenBank accession number XM_001846997.1) were designed and produced by Advanced Cell Diagnostics. Mosquito sections were selected based on preliminary images of uninfected *Culex pipiens* stained with haematoxylin and eosin. A negative control probe, supplied with the RNAscope^®^ 2.5 High Definition (HD)–Red Assay kit (ACD, Hayward, California, USA), was performed on parallel sections targeting the bacterial dihydrodipicolinate reductase (*DapB*) gene that was not expected to be present in the sample; any observed staining to this negative probe was due to non-specific binding. RNAscope^®^ staining procedures were conducted following the manufacturer’s instructions; all steps were performed at room temperatures with reagents supplied with the kit, unless stated. Slides were dewaxed for 5 min in xylene, immersed in 100% alcohol for 2 min, then dried. Slides were treated with hydrogen peroxide for 10 min, washed in water and placed into the target retrieval solution (supplied with the kit) for 15 min at 98–100 °C, then washed immediately with distilled water, followed by 100% alcohol and dried. A hydrophobic barrier was drawn around the section using the pen and templates provided and dried for 20 min or overnight. Sections were treated with Protease Plus (ACD) at 40 °C for 15 min.

Prepared sections were washed with water before applying the probes (RVFV, *GAPDH*, *DapB*) for 2 h at 40 °C and then washed in 1× wash buffer (twice for 2 min). After each of the subsequent hybridization steps, the sections were washed in 1× wash buffer (twice for 2 min). Solution AMP 1 (supplied by the manufacturer ACD) was applied for 30 min, followed by AMP 2 for 15 min, AMP 3 for 30 min and AMP 4 for 15 min, all at 40 °C. Buffers AMP 5 and 6 were incubated at room temperature for 30 and 15 min, respectively. Fast Red was then used to detect the signal by incubating at room temperature for 10 min. Slides were washed with water and counterstained with Gills 1 haematoxylin for 2 min and dried, either for 1 h at 60 °C or overnight at room temperature. A coverslip was applied using EcoMount (Biocare, California, USA). Images were captured with a Pannoramic 250 Flash II slide scanner (3D HISTECH, Budapest, Hungary), and image selection was performed on digitized images in a Pannoramic Viewer (Version 1.15.2 SP 2, 3D HISTECH).

## 3. Results and Discussion

To visualize virus infection within the tissues of *Culex pipiens*, mosquitoes (in groups of five) were fed a blood meal containing RVFV strain ZH501 and maintained for 7 days at 25 °C. Each mosquito was then fixed in formalin, and preliminary sections were stained with haematoxylin and eosin (Figure 1) to identify key structures.

To validate the RNAscope^®^ procedure and detect virus distribution, RNA probes were designed to detect a mosquito host target and RVFV virus genome. Sections were counterstained to highlight tissue structures. The negative control probe based on the DapB gene (Figure 2, left-hand panel) showed no staining, whereas the ubiquitously expressed host GAPDH probe showed extensive staining (Figure 2, right-hand panel). This confirmed the integrity of RNA within the tissue sample following the fixation and embedding process and that the probe detection method was successful.

Specific staining with the RVFV probe was detected in 20% of infected mosquitoes, similar to that observed in previous vector competence studies [12]. Whole mosquito sections stained with the RVFV probe demonstrated specific binding (Figure 3, right-hand panel). This detected RVFV RNA within the context of particular tissues within the mosquito body.

At higher magnification, the extent of RVFV replication could be assessed within individual tissues (Figure 4). Images of mock-infected sections are shown on the left-hand panel, whilst RVFV-infected images are shown on the right-hand panel. The principal observations are described below:

Midgut (Figure 4, panels 1–4). No staining was observed in the mock-infected mosquitoes. Rift Valley fever RNA was detected within the basal layer of the midgut epithelium, and extensive staining was observed in the proventriculus of the foregut. This suggests that by day 7, the virus had effectively crossed the midgut barrier [13]. The presence of RVFV infection of the proventriculus has been reported previously [14] and could represent a location for virus entry across the midgut epithelium.

Legs (Figure 4, panels 5 and 6). Rift Valley fever RNA was present within cross-sections of leg tissue from infected, but absent from mock-infected, mosquitoes. This confirms the utility of using leg samples to demonstrate virus dissemination in vector competence studies and suggests that there are cells capable of supporting virus replication within the leg structure.

Thorax (Figure 4, panels 7 and 8). Low levels of RVFV RNA were detected in the thoracic ganglia (G) and dorsal longitudinal muscles (DLM). This identifies potential virus replication sites in the close proximity to the salivary glands that were not located in these sections.

Head (Figure 4, panels 9 and 10). Rift Valley fever virus RNA was detected in the cerebral ganglion (G) relative to the rest of the head and thorax. Previous studies have suggested that RVFV infection of the ganglion could affect the mosquito regulatory functions [15] and, potentially, the behaviour of the infected mosquito.

Johnston’s organ (Figure 4, panels 11 and 12). The Johnston’s organ is found at the base of the antenna and detects vibration. Rift Valley fever virus replication was observed within the sensory scolopophores (Sc), with trace amounts of signal located at the cell bodies of the neurons (G).

Ommatidia (Figure 4, panels 13 and 14). Faint non-specific staining was observed on the cornea (CL) in the mock-infected section, but much stronger staining was observed in the RVFV-infected section, suggesting presence of RVFV RNA.

Ovarian follicles (Figure 4, panels 15 and 16). Rift Valley fever virus staining was observed on the exterior of the follicular epithelium but not within the follicles. This suggests that either the virus had not yet penetrated the follicle by day seven, or it is unable to enter the follicle of this species of mosquito. If the latter is the case, this tentatively implies that transovarial transmission does not occur. Further time-course investigations would be required to confirm this.

## 4. Conclusions

The results reported within this study complement those reported by [10], where L segment probes were used to detect RVFV in ruminant hepatic tissue as a diagnostic test. Here, we showed that a probe directed against the S segment designed to target the RVFV nucleoprotein-coding sequence detects RVFV RNA throughout a range of mosquito tissues, and this provides further evidence for the susceptibility of *Culex pipiens* to support virus escape from the midgut and replication at secondary sites, as recently reported by [12]. These data also provide proof of concept for the application of RNAscope^®^ in mosquito infection studies, demonstrating a novel highly sensitive method to detect virus RNA localization within the mosquito vector.

## Figures and Tables

**Figure 1 viruses-13-01079-f001:**
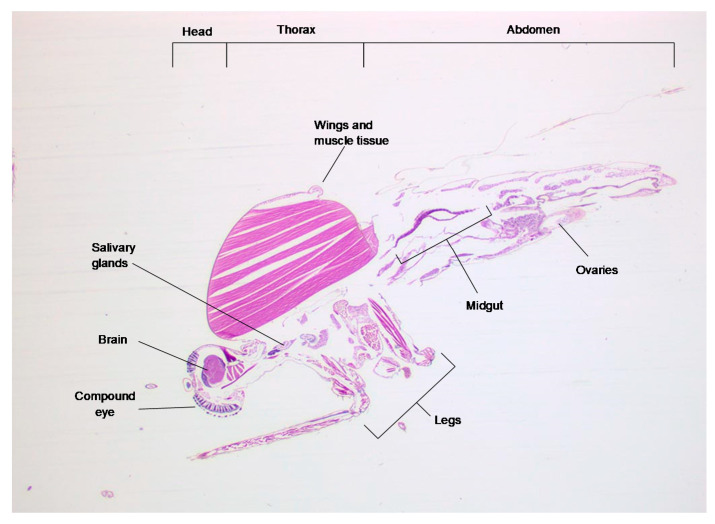
A haematoxylin and eosin stained 3 µm transverse section through an unfed *Culex pipiens* mosquito, showing the position of the main organ structures.

**Figure 2 viruses-13-01079-f002:**
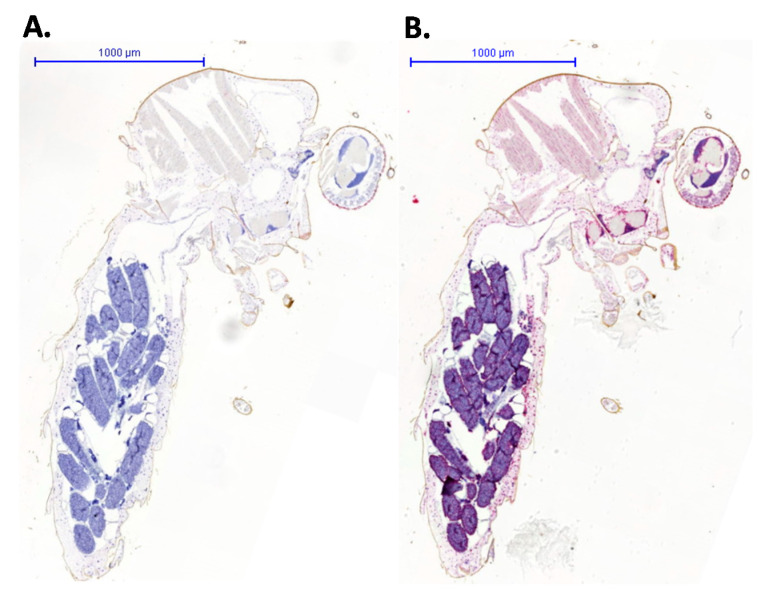
Whole *Culex pipiens* mosquito sections were stained with (**A**) a negative control probe that hybridized to the bacterial DapB RNA or (**B**) a positive control probe that bound to mosquito host GAPDH RNA. The bound probe was stained with Fast Red, and sections were counterstained with haematoxylin.

**Figure 3 viruses-13-01079-f003:**
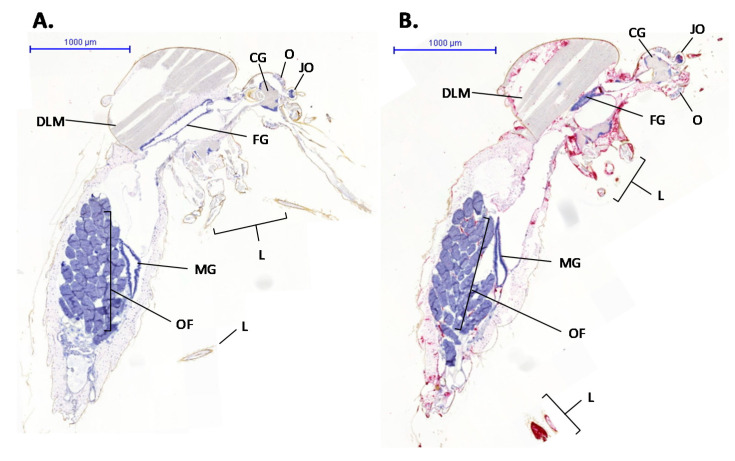
*Culex pipiens* mosquito sections treated with an RVFV-specific probe that binds to nucleoprotein RNA. Probe binding is shown by red staining. (**A**) A mosquito from the mock-infected control group. (**B**) An RVFV-infected mosquito. Organs labelled are CG, cerebral ganglion; JO, Johnston’s organ; O, ommatidia (compound eye); L, legs; FG, fore-gut; DLM, dorsal longitudinal muscle; MG, midgut; OF, ovarian follicle.

**Figure 4 viruses-13-01079-f004:**
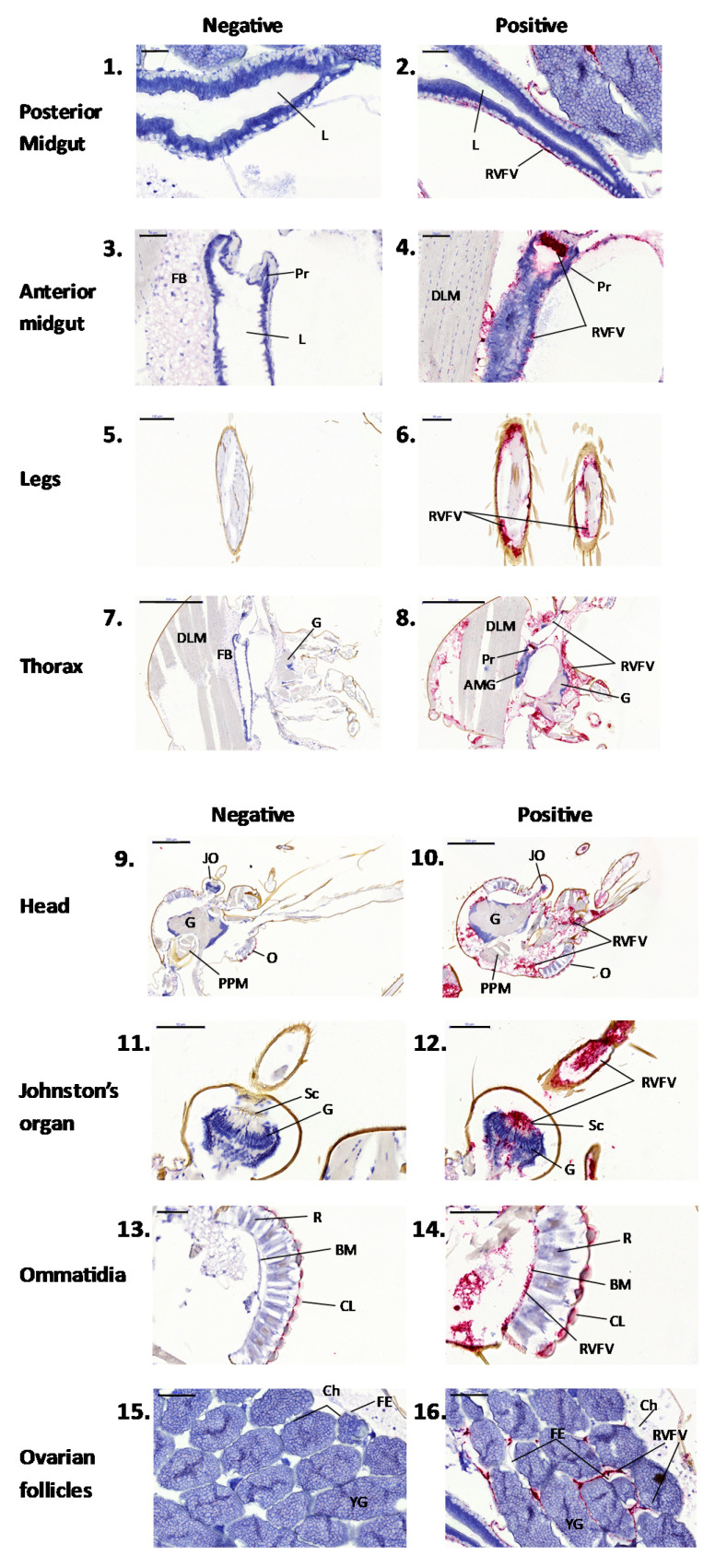
Higher magnification of tissues sections from Figure 2. Panels 1–4, midgut; panels 5 and 6, legs; panels 7 and 8, thorax; panels 9 and 10, head; panels 11 and 12, Johnston’s organ, including second segment of the antennae containing sensory cells; panels 13 and 14, ommatidia within the compound eye; panels 15 and 16, ovarian follicle. Abbreviations: AMG, anterior midgut; BM, basement membrane; Ch, chorion; CL, corneal lens; DLM, dorsal longitudinal muscle; FB, fat body, FE, follicular epithelium; G, ganglion JO, Johnson’s organ; L, lumen; O, ommatidia; PPM, pharyngeal pump muscle; Pr, proventriculus; OF, ovarian follicle; R, rhabdom surrounded by retinular cells; Sc, scolopophores; YG, yolk granules.

## Data Availability

All data has been presented in the article.

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
