# Peer review of "Detection of Rift Valley Fever Virus RNA in Formalin-Fixed Mosquitoes by In Situ Hybridization (RNAscope®)"

_viruses, 2021, doi:10.3390/v13061079_

Round 1
Reviewer 1 Report
The article submitted for my review demonstrates the very wide distribution of the Rift Valley Fever virus (RVFV) in different tissues of the Culex pipiens mosquito. The authors used an original in situ hybridization technique targeting viral RNA with the RNAScope method (which had previously been used to detect RFV virus in the liver of infected sheep); this study leads to the production of very beautiful images where the virus appears in red thanks to the staining by Fast red. The article thus validates the method for such investigative studies in mosquitoes infected with arboviruses and thus allow a better understanding of the virus-mosquito interaction so important in the control and prevention against arboviruses.
Some questions and remarks about this article:
- the authors correctly indicate that many species of mosquitoes are competent for RVFV; to do this, they cite a self-reference (reference 5) which lists competent English mosquitoes, but it would have been more appropriate to cite older studies for example which first showed the great variety of mosquitoes capable of multiplying the virus (as an example of articles by Turell MJ or more recently by Moutailler, 2008)
- reference 5 has been duplicated with reference 11 !
- rather than saying "as described by (11)", prefer the formula "as previously described (11)"
- line 65, specify the species which served for the blood meal infected with 107 viral units ; is it horse blood like the control group?
- explain why the authors highlight the fact that they used a probe derived from the S segment of RVFV when the pioneering study with the RNAScope technique had targeted a probe from the L segment ?
- in the description of the RNAScope technique, the abbreviation AMP is not explained, which for a non-user of the technique does not mean much; even though I searched the ACD site (the kit manufacturer), I did not find the meaning of this AMP
Author Response
Responses are indicated with >:
The article submitted for my review demonstrates the very wide distribution of the Rift Valley Fever virus (RVFV) in different tissues of the Culex pipiens mosquito. The authors used an original in situ hybridization technique targeting viral RNA with the RNAScope method (which had previously been used to detect RFV virus in the liver of infected sheep); this study leads to the production of very beautiful images where the virus appears in red thanks to the staining by Fast red. The article thus validates the method for such investigative studies in mosquitoes infected with arboviruses and thus allow a better understanding of the virus-mosquito interaction so important in the control and prevention against arboviruses.
>We thank the reviewer for the comment on the images.
Some questions and remarks about this article:
- the authors correctly indicate that many species of mosquitoes are competent for RVFV; to do this, they cite a self-reference (reference 5) which lists competent English mosquitoes, but it would have been more appropriate to cite older studies for example which first showed the great variety of mosquitoes capable of multiplying the virus (as an example of articles by Turell MJ or more recently by Moutailler, 2008)
> Reference 5 has been deleted and replaced with citations Turell et al 2008 and Moutailler et al 2008.
- reference 5 has been duplicated with reference 11 !
>deleted as stated above.
- rather than saying "as described by (11)", prefer the formula "as previously described (11)"
>the wording has been revised.
- line 65, specify the species which served for the blood meal infected with 107 viral units ; is it horse blood like the control group?
>Horse blood was used, the text has been modified to reflect this.
- explain why the authors highlight the fact that they used a probe derived from the S segment of RVFV when the pioneering study with the RNAScope technique had targeted a probe from the L segment ?
>The probe for this study was designed before the Ragan et al manuscript was published. We used the S segment as this was expected to bind to the virus nucleoprotein coding sequence , a sequence that is relatively conserved compared to other coding regions of negative-stranded viruses.
- in the description of the RNAScope technique, the abbreviation AMP is not explained, which for a non-user of the technique does not mean much; even though I searched the ACD site (the kit manufacturer), I did not find the meaning of this AMP
>AMP 1-6 are proprietary solutions supplied by the manufacturer. A comment has been added to this effect in the text.
Reviewer 2 Report
Detection of Rift Valley Fever Virus RNA in Formalin-fixed Mosquitoes by in situ Hybridization (RNAscope®)
Authors
Sarah Lumley , Laura Hunter , Kirsty Emery , Roger Hewson , Anthony R Fooks , Daniel L. Horton , Nicholas Johnson *
Abstract
Rift Valley fever virus (RVFV) causes a zoonotic mosquito-borne haemorrhagic disease that emerges to produce rapid large-scale outbreaks in livestock within sub-Saharan Africa. A range of mosquito species in Africa have been shown to transmit RVFV and recent studies have assessed whether temperate mosquito species are also capable of transmission. In order to support vector competence studies, the ability to visualize virus localization in mosquito cells and tissue would enhance the understanding of the infection process within the mosquito body. Here, the application of in situ hybridization utilizing RNAscope® to detect RVFV infection within the mosquito species Culex pipiens derived from the United Kingdom is demonstrated. Extensive RVFV replication was detected in many tissues of the mosquito with the notable exception of the interior of ovarian follicles.
In this brief report, the authors explain the application of specific ISH technology to visualize RVFV RNA encoding the NP in tissues using RNA scope methodology using mosquitoes infected with ZH501. This manuscript could benefit from some specific additions and modifications. The scientific soundness cannot be evaluated with the information provided. How many mosquitos were tested.? what is the resolution or the methods with regard to the vector and the current state of art; aka IHC and PCR?
Several specific recommendations for improvement of this manuscript are as follows.
Provide a better description of the ISH methodology that allows for the assessment of virus replication vs the presence of viral RNA. This terminology is interchanged in the manuscript and may not be the same thing depending on how the assay is constructed. This will also affect the conclusions that can be made using this technology moving forward with pathogenesis and transmission studies and thus the application of this methodology.
The number of mosquitoes used and tested is not provided and is necessary to access the scientific design of the study. see a few more comments about this below.
Line 34. Contaminate carcasses needs to be expanded to included aborted material. No mention of RVFV abortion in ruminants in the intro, this is very important aspect of this disease in humans and in animals
Line 42-45. The rationale and application or need for this method development could and should be greatly expanded here and in the conclusion; again this depends on this assay's ability to detect viral RNA or viral RNA associated with replication. How is ISH better or different than IHC used previously?
Localization to tissues and cells: the authors mention H and E staining ( hematoxylin and eosin) but there are no matching H and E’s. The inclusion of matching H and E would go far in helping visualize the organs and cells that they are trying to identify in the ISH images. This is essential for review and for this manuscript.
What is the origin of ZH501; what cells were used to propagate the virus used in this study and how might that affect the viral RNA distribution in the vector species? Why was this vector species chosen for this study?
Fig 1. They mention background staining in the eye ( panel A) this is very hard to visualize in this image- and insert may be warranted.
Fig 3 #13 and #14 again the background staining on the cornea is mentioned. The specific staining is much more intense. Maybe an asterisk/arrow would help identify the background staining for both these images.
Line 122 20% of how many? Seems low and only one reference is provided. Might want to expand on this. also a description regarding the difference in staining patterns or lack of it among various individuals evaluated with ISH is also warranted.
Fig 2 and 3. Please include matching and labeled H and E’s these are essential for organ/tissue/cell ID and for conclusions being made
In the results section, there are comments and conclusions made that although are interesting however several may be overreaching by the data presented here, perhaps moving some of these to the conclusions may be more appropriate. Such as the sentence in line141-142. 146-147. Both of these mention specific cells that can not be clearly ID in the ISH images; H and E would be helpful. Also, authors talk about a midgut barrier and the proventriculus as the traditional model of virus spread in the GI tract of this host, however, they do not talk about how the virus or its RNA gets to the leg cells ( what are these cells anyway) and how it gets to the ganglia or the DLM. This is by another mechanism? and could this other mechanism also allow for viral dissemination to the gut or salivary gland? Further clarification of the dissemination model and how these data are tied into it would be helpful to the reader to put the method's utility into scope. This is of particular importance when the model talks about the midgut barrier, virus dissemination to legs, and then to the salivary gland (how?) and subsequently into the saliva (which is coined as transmission-entomology term, not epidemiological terminology) . On that note there is no salivary gland is visualized in the manuscript; authors only mention that is not seen. There is a statement about organs testing positive within the vicinity; not sure of the relevance of this statement; is there a hypothesis that the virus gets to this organ by infection of proximity? or is the virus spread in the hemolymph?? The lack of images of the salivary glands seems to be a major limitation to the data set presented for the presence of virus in the gland and subsequently in the saliva is how this field evaluates the likelihood/capability of a vector species for virus transmission based on this current state of knowledge. This it is very important to see this organ staining positive at this stage of infection for previously these authors demonstrated that this vector species with ZH501 sheds live virus in saliva (ref#11). Demonstration of this level of resolution of this magnitude is necessary for proof of concept for use of ISH in mosquito infection/pathogenesis and transmission studies. Also, this virus reportedly kills mosquitoes; is the ISH staining different for those that scum to the infection?. Again this information would help demonstrate fit for purpose of ISH methodology in the vector. A sentence summarizing the resolution of the model with this methodology should replace the first sentence of the conclusion for the one that exists suggests the use of ISH for a diagnostic test for mosquitos and this application for most purposes is cost-prohibitive particularly given how affordable and ubiquitous PCR detection of many different pathogens in vectors has become.
Fig 3. Maybe break into two panels with large images; the current print version of these images is too small for the purpose.
Line 170/line 185 virus replication vs presence of viral RNA what is ISH detecting? This is of particular importance since it is one of the conclusions of the paper. How do you confirm virus replication at these locations? Or could it just be the presence of virus?
Ovarian follicles: H and E would be very helpful here. Line 178 is overreaching from ISH from one mosquito. It does not tentatively imply anything. It more aptly may e suggestive of limited transovarial transmission but much more extensive studies potently using ISH or IHC would need to be performed, these would need to include testing of live virus in eggs and infections in progeny.
185-189. midgut barrier again- followed by the proof of concept of ISH. See comments above. This should be further elaborated on how ISH can help in these studies particularly with regard to localization of viral RNA (and maybe virus replication) and the need and application in future research.
Author Response
Response to reviewers comments are indicated by >
In this brief report, the authors explain the application of specific ISH technology to visualize RVFV RNA encoding the NP in tissues using RNA scope methodology using mosquitoes infected with ZH501. This manuscript could benefit from some specific additions and modifications. The scientific soundness cannot be evaluated with the information provided. How many mosquitos were tested.? what is the resolution or the methods with regard to the vector and the current state of art; aka IHC and PCR?
>Within the manuscript we state, with references, the state of the art of IHC and ISH for studies detecting RVFV and how this brief report differs from them.
Several specific recommendations for improvement of this manuscript are as follows.
Provide a better description of the ISH methodology that allows for the assessment of virus replication vs the presence of viral RNA. This terminology is interchanged in the manuscript and may not be the same thing depending on how the assay is constructed. This will also affect the conclusions that can be made using this technology moving forward with pathogenesis and transmission studies and thus the application of this methodology.
>The manuscript describes in detail the methodology of this approach to visualizing RVFV RNA within infected mosquitoes. The terminology throughout the paper has been revised to acknowledge that this is its primary purpose.
The number of mosquitoes used and tested is not provided and is necessary to access the scientific design of the study. see a few more comments about this below.
>Details of the experimental approach are outlined in the methodology and cited reference (now 12). The procedure described in this manuscript reports preliminary findings for the processing individual mosquitoes and does not attempt to report a population level study.
Line 34. Contaminate carcasses needs to be expanded to included aborted material. No mention of RVFV abortion in ruminants in the intro, this is very important aspect of this disease in humans and in animals
>It is unclear why this is important to a study in mosquitoes or that abortion in humans is a very important aspect of RVFV. However, we acknowledge that abortion in ruminants is an important outcome of RVFV infection and have added text to this effect.
Line 42-45. The rationale and application or need for this method development could and should be greatly expanded here and in the conclusion; again this depends on this assay's ability to detect viral RNA or viral RNA associated with replication. How is ISH better or different than IHC used previously?
>The first benefit of ISH over IHC is its application on formalin-fixed material where some antibodies may not detect virus proteins due to the fixation process. Secondly, as the reviewer suggests above, ISH detects virus genome rather than expressed virus protein. The novel aspects of this brief communication are to report the application of this technique in mosquitoes and the use of a probe developed against the S segment.
Localization to tissues and cells: the authors mention H and E staining ( hematoxylin and eosin) but there are no matching H and E’s. The inclusion of matching H and E would go far in helping visualize the organs and cells that they are trying to identify in the ISH images. This is essential for review and for this manuscript.
>We disagree that the inclusion of hematoxylin and eosin images would add any value to this particular article. The images presented are counterstained with hematoxylin and provide equivalent detail to those with H & E staining.
What is the origin of ZH501; what cells were used to propagate the virus used in this study and how might that affect the viral RNA distribution in the vector species? Why was this vector species chosen for this study?
>Rift Valley fever virus strain ZH501 was originally isolated from a human infection in Egypt (state in the manuscript). The virus was propagated in Vero cells and had previously been used in a vector competence study in Culex pipiens. The use of Vero cells has been added to the manuscript in the Methods section. As a result, this vector species was used for the current investigation.
Fig 1. They mention background staining in the eye ( panel A) this is very hard to visualize in this image- and insert may be warranted.
>This sentence has been removed.
Fig 3 #13 and #14 again the background staining on the cornea is mentioned. The specific staining is much more intense. Maybe an asterisk/arrow would help identify the background staining for both these images.
>This text has been modified to reflect the contrast between the two sections.
Line 122 20% of how many? Seems low and only one reference is provided. Might want to expand on this. also a description regarding the difference in staining patterns or lack of it among various individuals evaluated with ISH is also warranted.
>This study has focused on the methodology and initial findings. Further studies are planned to apply both ISH and IHC on larger populations of mosquitoes to demonstrate variation over both time and between individuals.
Fig 2 and 3. Please include matching and labeled H and E’s these are essential for organ/tissue/cell ID and for conclusions being made
>As state above, the sections included in this article have been counterstained with haematoxylin and show the tissue structure of the mosquito.
In the results section, there are comments and conclusions made that although are interesting however several may be overreaching by the data presented here, perhaps moving some of these to the conclusions may be more appropriate. Such as the sentence in line141-142. 146-147. Both of these mention specific cells that can not be clearly ID in the ISH images; H and E would be helpful. Also, authors talk about a midgut barrier and the proventriculus as the traditional model of virus spread in the GI tract of this host, however, they do not talk about how the virus or its RNA gets to the leg cells ( what are these cells anyway) and how it gets to the ganglia or the DLM. This is by another mechanism? and could this other mechanism also allow for viral dissemination to the gut or salivary gland?
>The detection of RVFV RNA within the basal layer of the midgut provided sufficient evidence to include statements on potential routes but we agree with the reviewer that the current manuscript has clear limitations and that any conclusion drawn should be very tentative.
Further clarification of the dissemination model and how these data are tied into it would be helpful to the reader to put the method's utility into scope. This is of particular importance when the model talks about the midgut barrier, virus dissemination to legs, and then to the salivary gland (how?) and subsequently into the saliva (which is coined as transmission-entomology term, not epidemiological terminology) . On that note there is no salivary gland is visualized in the manuscript; authors only mention that is not seen. There is a statement about organs testing positive within the vicinity; not sure of the relevance of this statement; is there a hypothesis that the virus gets to this organ by infection of proximity? or is the virus spread in the hemolymph?? The lack of images of the salivary glands seems to be a major limitation to the data set presented for the presence of virus in the gland and subsequently in the saliva is how this field evaluates the likelihood/capability of a vector species for virus transmission based on this current state of knowledge. This it is very important to see this organ staining positive at this stage of infection for previously these authors demonstrated that this vector species with ZH501 sheds live virus in saliva (ref#11). Demonstration of this level of resolution of this magnitude is necessary for proof of concept for use of ISH in mosquito infection/pathogenesis and transmission studies. Also, this virus reportedly kills mosquitoes; is the ISH staining different for those that scum to the infection?. Again this information would help demonstrate fit for purpose of ISH methodology in the vector. A sentence summarizing the resolution of the model with this methodology should replace the first sentence of the conclusion for the one that exists suggests the use of ISH for a diagnostic test for mosquitos and this application for most purposes is cost-prohibitive particularly given how affordable and ubiquitous PCR detection of many different pathogens in vectors has become.
Fig 3. Maybe break into two panels with large images; the current print version of these images is too small for the purpose.
>As we acknowledge in the manuscript, we were unable to identify the salivary glands and this is clearly a limitation of this study in attempting to understand the transmission of RVFV by infected mosquitoes. However, we believe that the study provides sufficient controls to demonstrate the presence of RVFV RNA within infected mosquitoes and that technique as described could be a useful tool in understanding virus infection in the vector. Due to these limitations we do not attempt to speculate on the dynamics of RVFV infection once the virus has crossed the midgut barrier. Presumably, dissemination occurs through the haemolymph, although we do not state this in the text, accessing a wide range of tissues that could become infected, including cells within the legs as suggested by our images (Fig 3, panel 6).
Line 170/line 185 virus replication vs presence of viral RNA what is ISH detecting? This is of particular importance since it is one of the conclusions of the paper. How do you confirm virus replication at these locations? Or could it just be the presence of virus?
>As agreed above, we limit reference to RVFV RNA unless highly qualified.
Ovarian follicles: H and E would be very helpful here. Line 178 is overreaching from ISH from one mosquito. It does not tentatively imply anything. It more aptly may e suggestive of limited transovarial transmission but much more extensive studies potently using ISH or IHC would need to be performed, these would need to include testing of live virus in eggs and infections in progeny.
>We agree that further studies are required and limit the speculation in this brief article.
185-189. midgut barrier again- followed by the proof of concept of ISH. See comments above. This should be further elaborated on how ISH can help in these studies particularly with regard to localization of viral RNA (and maybe virus replication) and the need and application in future research.
>As stated in the introduction, the benefit of ISH is the detection of RVFV genome rather than protein, and its application with formalin-fixed samples where antibodies are not effective.
Round 2
Reviewer 2 Report
The inclusion of H&E images would greatly strengthen this paper's quality and scientific conclusions by allowing morphological gold standard localization of specific staining.
Does ISH use a sense or antisense probe? this is related to what is actually being detected in this methodology and the conclusions that can be made in future applications of this methodology.
Again numbers of animals used in the protocol development are not provided. given how important it is to demonstrate staining in the salivary gland. How hard is it to get the salivary gland in a section or is it there and just not staining? if it's that challenging that it can not be included in the brief methods review to demonstrate the utility of the said method in pathogenesis research, then one is left wondering how the methods could be improved for more precise sectioning and the inclusion of H&E for organ identification.
Author Response
We thank the reviewer again for challenging us to improve our brief manuscript. In the revision we have added an additional figure showing a H&E stained section of an unfed mosquito to illustrate mosquito anatomy. A comment have been added to state that the target for the probe was the negative-stranded RVFV RNA , thus the procedure detects virus genome.